# Enabling Semantic-Functional Communications for Multiuser Event Transmissions via Wireless Power Transfer

**DOI:** 10.3390/s23052707

**Published:** 2023-03-01

**Authors:** Pedro E. Gória Silva, Nicola Marchetti, Pedro H. J. Nardelli, Rausley A. A. de Souza

**Affiliations:** 1School of Energy Systems, Lappeenranta–Lahti University of Technology (LUT), 53850 Lappeenranta, Finland; 2Department of Electrical Engineering, National Institute of Telecommunications (INATEL), Santa Rita do Sapucaí 37540-000, Brazil; 3Connect Centre, Trinity College Dublin, D02 PN40 Dublin, Ireland; 46G Flagship, University of Oulu, 90570 Oulu, Finland

**Keywords:** wireless power transfer, energy harvesting, semantic-functional communications, event-based communications

## Abstract

A central concern for large-scale sensor networks and the Internet of Things (IoT) has been
battery capacity and how to recharge it. Recent advances have pointed to a technique capable of
collecting energy from radio frequency (RF) waves called radio frequency-based energy harvesting
(RF-EH) as a solution for low-power networks where cables or even changing the battery is unfeasible.
The technical literature addresses energy harvesting techniques as an isolated block by dealing with
energy harvesting apart from the other aspects inherent to the transmitter and receiver. Thus,
the energy spent on data transmission cannot be used together to charge the battery and decode
information. As an extension to them, we propose here a method that enables the information to
be recovered from the battery charge by designing a sensor network operating with a semanticfunctional
communication framework. Moreover, we propose an event-driven sensor network
in which batteries are recharged by applying the technique RF-EH. In order to evaluate system
performance, we investigated event signaling, event detection, empty battery, and signaling success
rates, as well as the Age of Information (AoI). We discuss how the main parameters are related to the
system behavior based on a representative case study, also discussing the battery charge behavior.
Numerical results corroborate the effectiveness of the proposed system.

## 1. Introduction

Between the end of the 19th century and the beginning of the 20th century, Nikola Tesla envisioned, designed, and built an experimental station (Wardenclyffe Tower) to transmit energy over the air. Tesla’s technique applied alternating electrical current to the ground so that a device touching the ground would close an electrical circuit through the atmosphere with the generator, and the gadget would thus be powered. Although Tesla did not address the use of radio waves, one can understand Tesla’s experiment, the Wardenclyffe Tower, as a historic milestone initiating the radio frequency-based energy harvesting (RF-EH) field of research. Following the trend of miniaturization of Internet of Things (IoT) devices, including sensors, research focused on wireless charging devices at a distance has recently gained increasing relevance.

Wireless power transfer, a broader term than RF-EH, can be separated into two primary categories: near-field and far-field. This sub-classification arrives from the distinct behavior of electromagnetic waves due to the distance from the transmitting antenna. Overall, the area delimited by one wavelength from the electromagnetic wave source constitutes the near field. In general, this distance is no more than a few meters, thus drastically limiting the coverage area for possibleWireless power transfer harvesting (WPT) techniques with a near-field application; on the other hand, far-field methods are capable of covering a vast area at the cost of providing the device with reduced power compared to near-field ones [1,2,3,4].

In addition to energy solutions, one can highlight two other aspects relevant to the design of a system: Application and Communication. For simplicity, we group all works that address concepts related to a specific function, general objective, highest communication layer (e.g., Open Systems Interconnection (OSI) model), etc., inside the Application. In a way, the Application contemplates solutions solely dealing with the primary function of operating the system. The Communication umbrella covers works whose purpose is to assess the performance of information exchange; therefore, it contains methods, techniques, and analyses that study modulation performance, coding, channel capacity, different medium access control techniques, etc. Finally, we can add another class that includes work related to energy consumption, named Energy here. Energy refers to work that aims to understand and evaluate the system’s energy consumption or expenditure.

An outline of how literature deals with these classes is described in Figure 1 together with Table 1. Note that there are intersection regions between the subsets in Figure 1 representing works that directly address two or three aspects listed above. Table 1 is directly related to Figure 1; each subset of Figure 1 (named #1, #2, #3, etc.) is exemplified through one or more papers from the literature in Table 1. Although one can extend the list of examples to subsets #1 through #6, to the best of the authors’ knowledge, this is the first work to jointly consider aspects of Energy, Application, and Communication in the system design. Thus, in addition to contrasting with other works, it is not possible to draw a parallel or fairly compare our solution with other methods presented in the technical literature. In order to exemplify such a discrepancy, suppose a comparison with a Media Access Control (MAC) protocol. In this case, any clarification possibly acquired would refer exclusively to Communication, regardless of the specific method adopted (Time-Division Multiple Access (TDMA), Frequency-Division Multiple Access (FDMA), Code-Division Multiple Access (CDMA), etc.). Furthermore, one could propose a system covering the three classes (Energy, Application, and Communication) consisting of methods designed or evaluated specifically for one or two (never for all three) classes as a benchmark. This case would not be an adequate comparison either; as will become clear throughout this paper, the proposed solution jointly covers the three classes and cannot be understood as a combination of three dispersed methods.

This paper contributes to the literature in the field (which will be reviewed in the sections to come; a complete landscape can be found in [14]) by combining wireless power transfer techniques and the semantic-function approach recently proposed in [12] that consider meaningful events that require explicit use of the communications channel to support the operation of a given cyber-physical system [15]. Our numerical results demonstrated that it is possible to have an effective recharge of batteries and optimized performance by merging RF-EH and the semantic-function approach.

The rest of this paper is divided as follows: Preliminary concepts are introduced in Section 2. The problem addressed is formulated in Section 3. The proposed system is introduced in Section 4. A case study is carried out in Section 5 in order to assess the performance of the system through simulations. Then, we present the conclusions of the paper in Section 6.

## 2. Background: WPT and Fundamentals of RF-EH

In this section, we will provide a brief overview of wireless power transfer and RF-based energy harvesting that will serve as the main theoretical background of this paper.

### 2.1. Near-Field WPT

The near-field or non-radiative region is the zone within one wavelength from the transmitting antenna. Inside the near-field area, one can power a device with up to a few tens of Watts by employing inductive coupling, resonant inductive coupling, or capacitive coupling in the range of tenths of Watts [4,16,17]. The near-field solutions, under certain conditions, reach more than 90% efficiency in the energy transfer process between the source and the powered device [17]. An advantageous feature of near-field methods is that power dissipation only occurs when we have an absorbing material (a device to be charged) within the actuation component. In other words, the transmitter in a near-field method drastically reduces its energy expenditure (energy consumed for its operation in addition to transfers when applicable) if there is no device to charge. This peculiarity provides an extra monitoring capacity in the system: the transmitter can identify the presence of nearby devices through its energy consumption. On the other hand, near-field techniques face an exponential decrease in the total power transmitted by expanding the gap between the source and load. Overall, we have smallish or no energy transfer when the distance between the source and load exceeds one wavelength.

#### 2.1.1. Inductive Coupling

In inductive energy transfers for near-field WPT, the power transmitter recharges the devices through a magnetic field induced by coils. In this type of application, efficiency is as high as the proximity of the devices [18]. Inductive power transfer is supported theoretically by Ampere’s law: an alternating electric current produces an oscillating magnetic field when passing through a coil or wire. Moreover, once the magnetic field generated by the power transmitter is close enough to the receiver coil, we have an alternating electric current in the receiver; then, the receiver recharges its battery as soon as it rectifies the obtained alternating current. A wide range of applications uses this technique. We can mention electric toothbrush stands at low frequencies (50/60 Hz) or consumer electronic devices such as laptops, cellular phones, and other portable devices, in addition to those used to charge electric automobiles at high frequencies [1]. In order to enhance inductive power transmission, one can add resonant circuits. In resonant inductive power transfer (RIPT), resonant circuits consist of a coil of wire connected to a capacitor or a resonator with internal capacitance. Like inductive power transfer techniques, magnetic fields convey energy from the transmitter to the receiver in RIPT; however, WPT based on resonant inductive power transfer can cover broader areas. In addition, there are several advantages of resonant RIPT over inductive power transfer, such as reduced electromagnetic interference, higher frequency of operation, and higher efficiency. Among the possible applications for RIPT, we can mention wireless power coverage, power lights, and recharging mobile batteries anywhere in a room without wired connections.

#### 2.1.2. Capacitive Coupling

In capacitive energy transfers for near-field WPT, the power transmitter recharges the devices through an electric field between electrodes like metal plates. More precisely, the transmitting power source applies an alternating voltage across a metal plate, leading to electrostatic induction and an alternating electrostatic potential on the metal plates of the receiver. This receiver’s metal plate is the alternating current source to recharge the device. Some fundamental factors for the efficiency of energy transfer are frequency, the square of the voltage, and the capacitance between the transmitter and receiver plates [19]. Another intriguing characteristic of capacitive coupling is the proportionality between the transferred power and the smallest area among the plates used in the system. Capacitive energy transfers involve limited power due to poisonous ozone gas produced by high voltage on the electrodes. Therefore, techniques of capacitive energy transfer are applied to low-power systems due to their hazardous property. However, capacitive couplings have advantages over inductive couplings, such as controlled interference, and the alignment between the transmitter and receiver can be more flexible. Recent work has pointed to capacitive coupling as a potential method for an electric double layer for wireless power transfer under seawater [20].

### 2.2. Far-Field WPT

The far-field or radiative region is the zone beyond one wavelength around the transmitting antenna. In this region, the electromagnetic field settles into ordinary electromagnetic radiation, i.e., we have transverse electric or magnetic fields with electric dipole characteristics. In addition, the radiated power is well-behaved, decreasing with the square of the distance, and the radiation absorption does not feed back to the transmitter. We can point out radio waves, microwaves, and laser beams as examples of electromagnetic radiations that are traditionally adopted for far-field WPT. In general, radio waves have a lower attenuation and are more efficient than microwaves or laser beams due to atmospheric attenuation. Given the intrinsic scattering of propagation of electromagnetic waves, as a rule of thumb, the efficiency of a far-field WPT does not exceed 50% [21]. On the other hand, for example, as long as there is a line-of-sight propagation, Ref. [22] has shown that more than 80% efficiency can be achieved.

The literature usually employs far-field WPT techniques to cover a wide area or charge devices far from the source. Some available power sources may not be dedicated to recharging devices and still be appropriated for the far-field WPT technique. In other words, a timely use can be made of the available frequency spectrum by harvesting the power traveling freely in the air. Therefore, it is reasonable to classify power sources into two classes: dedicated power sources and ambient power sources. Dedicated power sources can, for example, benefit from a licensed frequency band or even a license-free band to charge a remote device. On the other hand, an ambient power source is any transmitter whose transmitted electromagnetic waves reach the device. Although this distinction defines the power sources, it clarifies relevant aspects regarding the receiver design. In general, receivers that collect energy from ambient power sources must consider several medium statistics, while scenarios with dedicated sources are less susceptible to the environment. We shall see this in more detail throughout this work.

The literature has lately pointed to RF-EH as a disruptive technology that enables low-power portable devices and energy-constrained wireless networks to benefit from the electromagnetic energy available in the environment. Roughly speaking, the working principle of RF-EH consists of an electric-to-electromagnetic-to-electric conversion process in order to recharge wireless devices. It should be noted that the first process transformation, electric-electromagnetic, is not necessarily carried out by a source dedicated to the RF-EH system. Researchers have proposed several architectures to perform energy harvesting in order to coexist with traditional data transmission systems. Briefly, although there are different architectures for RF-EH, to the best of the authors’ knowledge, the transmission method employed does not deviate from the more traditional modulation techniques. Figure 2 illustrates the main architectures for RF-EH regarding the arrangement of the data layer concerning the energy harvesting layer. We discuss each of them in more detail below.

#### 2.2.1. Separate Architecture

As highlighted in Figure 2a, the separate architecture consists of two or more antennas and two independent receiver blocks. Each receiver block performs a different function: data decoding (data recovery) and electromagnetic wave conversion into the direct current (energy harvesting). More than one receiving antenna can be used for each block, depending on the technique employed. For example, multiple-input-multiple-output (MIMO) can be used for the data transmission system, or one can have antennas in different bands to harvest energy from various frequencies. Another example is [23], in which the authors propose a MIMO broadcast system for simultaneous wireless information and power transfer. In this type of architecture, the receiver can perform both the energy harvesting function and the data recovery continuously and concurrently.

#### 2.2.2. Time Switching Architecture

A Time Switching architecture must have an antenna selector switch, a data receiver, and an energy harvesting block, as shown in Figure 2b. It is commonly assumed that all antennas feed exclusively one block at a time. The antenna selector switch can modify the receiving process of the system, i.e., it can select either energy harvesting operation or data recovery. In a nutshell, the working principle of time switching consists of periodically switching the beam between energy harvesting and data recovery blocks. Depending on the adopted data transmission method, it may be necessary to synchronize the operation mode selection. In other words, the receiver may not be able to receive the transmitted data because it is operating in energy harvesting mode. On the other hand, this architecture provides the highest peak value for the harvested energy compared to other architectures with equivalent systems. This advantage can be quite beneficial in environments with sporadic availability of energy. Ref. [24] studies the optimal design for simultaneous wireless information and power transfer in down-link orthogonal frequency-division multiplexing (OFDM) systems, in which the users apply either time switching or power splitting to coordinate the energy harvesting and data recovery processes.

#### 2.2.3. Antenna Switching Architecture

A sketch of the antenna switching architecture is shown in Figure 2c. This architecture is constituted by two independent receiver blocks and two or more antennas. We have one data recovery block and an energy harvesting one. The Antenna Switching architecture is distinguished by using a low-complexity switch. Antennas can migrate from one block to the other under certain conditions. As the dynamic properties of antenna systems do not require high speed or periodic changes in the configuration of the antenna array of each block, one can claim that antenna switching involves a relatively low complexity and that this architecture is, somehow, more suitable for practical designs [1].

#### 2.2.4. Power Splitting Architecture

The power splitting architecture has a peculiar characteristic: the division of the power collected by all the antennas into two streams of different power levels before the receiver performs signal processing. As with the separate architecture, the power splitting one operates with two independent receiver blocks. We have one receive block to recover the information and another to harvest energy, as depicted in Figure 2d. The number of antennas may vary according to the application and the frequency range of interest. This architecture allows the receiver to perform both the energy harvesting function and the data recovery continuously and concurrently. Some approaches, such as those proposed by [25], present the division of power between energy harvesting and data recovery blocks as dynamic. This characteristic is a significant advantage of this architecture, as it allows the receiver to operate with different reception rates according to the need for a recharge. Furthermore, one can design the power-splitting block like any other power splitter of conventional communication systems [26]. We can find a vast literature on applications to power division architectures, such as [27], which proposes a long-range optical wireless energy and information transfer system of high power and high rate.

### 2.3. RF Energy Sources and Statistics

Following the classification proposed earlier in this paper, Figure 3 summarizes the energy sources available to harvest energy. The first branch in Figure 3 concerns the energy emitter. As already discussed, sources emitting energy without the specific purpose of recharging a device—this certainly includes the forces of nature—are treated as ambient sources, and transmitters of electromagnetic waves to power a remote device are classified as dedicated sources.

We split ambient sources into sub-classes related to the nature of the energetic process. Photovoltaic covers sources that can be harvested by converting light into direct current; therefore, any light source belongs to the Photovoltaic class. Miscellaneous sources of electromagnetic waves outside the frequency of visible light are classified as RF class since they are not dedicated to power devices. For cases in which the energy harvesting process consists of a transformation of mechanical energy into electrical energy, we classify them as Flow; two sub-classes are highlighted for it: Wind and Hydro. The last class of Ambient Sources is related to thermal energies. For example, wearable sensors which can recharge themselves using human heat are classified as Thermal. Applications in the Thermal class require the device to handle thermoelectric phenomena.

Dedicated sources can handle the most diverse means of delivering power to the remote device. We could divide it into the same categories proposed for Ambient Sources; however, by and large, the literature suggests electromagnetic wave transmitters as the dedicated source. We then split the Dedicated RF class into Continuous and On Demand. The former refers to sources that operate and transmit uninterruptedly—note that this classification does not restrict the transmitter regarding the operating mode, i.e., the transmitter can transmit information continuously. The latter implies that the transmitter has some control and only transmits energy at opportune times, even if transmissions are periodic.

Since our focus in this work is RF-EH, we shall approach RF from Ambient Sources and Dedicated RF classes. The reader is referred to [2] to obtain further insights about other energy sources.

### 2.4. RF from Ambient Sources

We can highlight radio broadcasts, TV broadcasts, Wi-Fi, and mobile phone networks as the most typical sources for RF-EH in urban centers. Except for specific cases, such as the receiver being extremely close to a base station, the power available after harvesting from these sources does not exceed tens of mW. Due to this low power, the rectifier circuits and matching network can reach an efficiency of 70% at best [2]. However, an energy conversion efficiency of around 83% can be achieved, under certain conditions, by optimizing the antenna array and the rectifier circuit, employing the gradient method [28]. Due to the nature of the medium and sources, the amount of energy captured by the RF-EH method from ambient RF sources randomly varies over time. Some studies have focused on modeling several frequency ranges in a probability density function. For instance, a kernel distribution model is used to fit experimentally collected data on power density in some RF bands in [29]. Furthermore, the authors compare the data with the Rayleigh distribution. Ref. [30] assesses the broadband RF radiations in terms of power densities from 467.5 MHz to 3.5 GHz, in addition to graphically presenting the probability density function (pdf) and cumulative density function (cdf) of the power densities. Ref. [30] does not propose a theoretical pdf for the collected data; however, the authors indicate the mean and standard deviation of the samples. Since a trustworthy characterization of the medium is beyond the scope of this work, the knowledge available at present in the literature about random variations over time of the medium is sufficient for our exposition. In particular, we assume the following mathematical statement to characterize the power harvested from ambient sources.

**Assumption 1.** 
*Given an RF-EH receiver with efficiency η, the energy harvested from Ambient Sources after T seconds is given by*

(1)
$$E_{\mathrm{Ab}}\left(T\right)=\eta \int_{T}E_{\mathrm{Ab}}\left(t\right)dt=T\eta p_{\mathrm{Ab}},$$

*with $E_{\mathrm{Ab}}\left(t\right)$ representing the the instantaneous energy density at the RF-EH receiver, and $p_{\mathrm{Ab}}$ follows a Rayleigh distribution as*

(2)
$$f\left(p_{\mathrm{Ab}}\right)=\frac{p_{\mathrm{Ab}}}{\sigma_{\mathrm{Ab}}^{2}}\exp\left(-\frac{p_{\mathrm{Ab}}^{2}}{\sigma_{\mathrm{Ab}}^{2}}\right).$$

*The mean of $p_{\mathrm{Ab}}$ is given by $\sigma_{\mathrm{Ab}}\sqrt{\pi /2}$.*


### 2.5. Dedicated RF

As already discussed previously, we named as dedicated those RF sources that intentionally deliver energy in one or more RF frequency ranges to wireless sensor nodes or mobile devices. These sources are fully controllable, and thus, they are predictable. On the other hand, the system’s unpredictability arises from the communication channel. Let us examine this issue more closely.

For a multipath channel, we have that the complex envelope of the received signal is given by [31]
(3)r(t)=x(t)h(t)+n(t),
with x(t), h(t), and n(t) being the complex envelope of the transmitted signal, channel coefficient, and additive white Gaussian noise (AWGN), respectively. Thereby, the RF-EH can harvest a certain amount of energy at the end of an exposition period *T* given by
(4)$$E_{\mathrm{De}}\left(T\right)=\eta \int_{T}{\left(r\left(t\right)\right)}^{2}\;dt.$$

For an unmodulated carrier $x\left(t\right)=A_{r}$ and slow fading, and assuming a Rician fading channel, we have that energy harvested by RF-EH receiver is given by
(5)$$E_{\mathrm{De}}\left(T\right)=\eta \int_{T}{\left(A_{r}h\left(t\right)+n\left(t\right)\right)}^{2}dt=\eta \left(T{\left(A_{r}h\right)}^{2}+2A_{r}hn+\int_{T}n{\left(t\right)}^{2}dt\right),$$
in which *n* is zero-mean, $\sigma^{2}$-variance Gaussian variate, and *h* follows a Rician distribution with mean and variance given by $E\left[h\right]=H_{m}$ and $\mathrm{VAR}\left[h\right]=\sigma_{\mathrm{De}}$, respectively.

Precisely determining the pdf of $E_{\mathrm{De}}\left(T\right)$ is beyond the scope of this work. However, given the intrinsic characteristics of the RF-EH process (antenna, impedance matching cell, rectifier circuit, etc.), we assume that the RF-EH receiver’s bandwidth is narrow enough such that n(t) can be considered constant over a period of *T* seconds. Accordingly, we can rewrite (Equation 5) as
(6)$$E_{\mathrm{De}}\left(T\right)=\eta T\left({\left(A_{r}h\right)}^{2}+2A_{r}hn+n^{2}\right).$$

Figure 4 presents three histogram examples for the energy harvested by RF-EH receiver $E_{\mathrm{De}}\left(T\right)$. We have T=1, η=1, $A_{r}=1$, and $\sigma^{2}=0.01$ in all three histograms. A Rician fading with means of 2, 0.77, and 1 and variance of 0.1, 0.15, and 0.01 were assumed in blue, orange, and yellow, respectively.

## 3. Problem Formulation

This section introduces the considered scenario and our system model. We discuss how our approach differs from traditional solutions by briefly presenting how the literature approaches the problem. Furthermore, we discuss how the sensors can harvest energy and receive information without jeopardizing the system’s operation.

Feedback control demands communications as a way of assessing the current state of a physical entity. The feedback control design of any remote physical process must cover two main aspects: control theory and communication theory. From a control theory perspective, the decision-making unit (data fusion or control center) acquires information about the physical process through the sensor network, in order to predict which intervention should be performed by the physical entity. The key challenges are how and when to act (controller block) and what information to acquire from the physical system (sensors and feedback). By receiving meaningful events instead of periodic samples, the decision-maker can control the system adequately [32]. This approach is called event-triggered control (ETC) and may save network resources while ensuring system stability [7]. Concerning communication aspects, one can mention a Wireless Sensor Network (WSN) as a case of interest here. In a nutshell, a WSN consists of sensors that operate remotely and report their readings to the decision-maker through a wireless channel. For a WSN, event-driven communication
(EDC), which consists of transmitting only when certain events occur, can work in harmony with ETC. The distinction between EDC and ETC is subtle, although they address different system parts—the former handles the communication layer, and the latter addresses the control part of the system.

The convergence of EDC and ETC in a specific event-driven communication system designed to perform event-triggered control is still an open problem. The work we proposed in [12], to the best of the author’s knowledge, is the first to merge control and communication aspects in a dedicated communication system, taking advantage of implicit communications between transmitters (sensors) and receivers (controllers). The approach proposed in [12] determines the event (sensor that transmitted it) based on an identification code generated by a random map. Since the above-mentioned technique works with energy detection instead of a strict signal demodulation process, we will use it in conjunction with RF-EH in order to ensure the operation of a monitoring system with sensors powered by RF. More details about the system and how it works are presented below.

## 4. System Model

Our scenario consists of $N_{s}$ sensors spread over an area. This scenario can represent, for example, an electric power plant, an industry, a rural area, a farm, or any other environment that requires monitoring. In addition to the sensors, there are also $N_{u}$ unmanned aerial
vehicles (UAVs) randomly flying over the region in order to collect information from the sensors and forward it to the decision-maker. For comparison purposes, we also assume a scenario where the area is small enough so that the sensors communicate directly with the decision-maker. This work aims to investigate the behavior of the sensors regarding the battery charge and how they access the medium. Therefore, we consider an ideal channel between the UAVs and the decision-maker. Furthermore, we do not assess the control system’s performance in terms of stability; only sensor network properties such as delay are investigated. We also assume that the system employs distinct frequency bands for Dedicated RF and Ambient Sources harvesting.

An operation diagram of sensor nodes is presented in Figure 5. The sensors remain in an idle state (deep sleep) as long as an event does not occur. Deep sleep is a widely used operation in embedded systems to save energy. In this state, micro-controllers and other components can operate under significantly low electrical currents, around 20 μA [33]. In addition, one can wake up the micro-controller due to external factors, in our case, as the result of an event. The rise of a signal (which can be, for example, the measure of temperature, pressure, or humidity) above a threshold may trigger what we treat as an event; more precisely, one can define a function event as general as possible as f:R→{0,1}, such that f(t)=1 represents the event’s occurrence (the reader interested in more details about the event is referred to [12]). In other words, given a signal of interest, the event can be defined as required by the decision-maker. After identifying the event, the sensor node starts the transmission, which we discuss in more detail later. Immediately after the the end of the transmission, the sensor node saves the battery charge and accurately monitors the current from the RF-EH circuit. After *W* seconds of current monitoring, the sensor node decides whether to retransmit based on the behavior of the battery charge current over the last W seconds. Note that the acknowledgment (ACK) transmitted by one or more UAVs is not a conventional data packet; instead, our ACK is identified through the battery charging pattern in the sensor. Thus, the ACK simultaneously fulfills two roles: recharging the battery and confirming receipt of the message. Sporadic conditions, such as insufficient battery power, bring the sensor node back to deep sleep.

Communication between sensors and UAVs in both directions is established through the semantic-functional communication (SFC) technique proposed in [12]. In a nutshell, each event obtains a sole transmission map that enables unambiguously determining the origin and the event at the receiver for SFC. This map guides the transmission of energy packets over time, called energy slots—to the best of the authors’ knowledge, the concept of energy slot proposed in [12] differs from standard transmission techniques by not requiring a process of demodulation itself in addition to a distinctive methodology for controlling access to the medium; therefore, one should not understand it as a traditional method of transmission.

In our scenario, the UAV listens to the sensors through the SFC and then replies to them with an ACK again using the SFC. Thus, the UAV must look for patterns of energy variation over time in order to recover the transmitted information. Here, we will not thoroughly describe the UAV reception process; it suffices to say that SFC enables the UAV to always identify, under certain conditions, an event at the cost of a low rate (in many cases negligible) of false alarms. On the other hand, a sensor may be out of range of the UAV at a given time, leading to a failure in communication. We will assume that sensor transmissions have a success rate of $R_{s}$ proportional to the number of UAVs per area.

The reception process at the sensor consists of determining if it hears an ACK transmission by scrutinizing the battery charging as follows. Let $T_{E}$ and $T_{E}^{\left(k\right)}=\left\{t\in R:\left(k-1\right)T_{E}<t-t_{\mathrm{TE}}^{\left(n\right)}\leq kT_{E}\right\}$ be the duration of an energy slot and the time interval of the *k*th energy slot just after sensor transmission, respectively, with $t_{\mathrm{TE}}^{\left(n\right)}$ being the time at which the *n*th transmission ended. Thus, after transmitting, the sensor node samples the charge (energy) of the battery every $T_{E}$ seconds—for simplicity, we assume that this process is perfect, i.e., the amount of energy in a given instant is unequivocally read by the sensor. Let the battery charge at time *t* in the sensor S be given by $C_{S}\left(t\right)\leq C_{\max}$. Thus, at the instant just after transmission, we have that the change in battery charge is given by
(7)$$\begin{array}{cc} \Delta C[k] & =C_{S}\left(kT_{E}+t_{\mathrm{TE}}^{\left(n\right)}\right)-C_{S}\left(\left(k-1\right)T_{E}+t_{\mathrm{TE}}^{\left(n\right)}\right)\\ \; & =P_{\mathrm{De}}\left[k\right]E_{\mathrm{De}}\left(T_{E}^{\left(k\right)}\right)+E_{\mathrm{Ab}}\left(T_{E}^{\left(k\right)}\right)\;\forall k\in \left\{1,2,3,\cdots ,K\right\} \end{array}$$
where
(8)$$P_{\mathrm{De}}\left[k\right]=\left\{\begin{array}{cc} 1, & \mathrm{if}\;\mathrm{UAV}\;\mathrm{transmitted}\;\mathrm{in}\;T_{E}^{\left(k\right)}\\ 0, & \mathrm{otherwise} \end{array}\right.\;,$$
and K≤⌊W/T⌋, with ⌊·⌋ rounding to the largest integer less than or equal to the argument.

Let the threshold function f:R→{0,1} be given by
(9)$$f\left(a\right)=\left\{\begin{array}{cc} 1, & \mathrm{if}\;a>\delta \\ 0, & \mathrm{otherwise} \end{array}\right.\;.$$

Then, by making $B_{C}\left[k\right]=f\left(\Delta C\left[k\right]\right)$, the sensor node has a binary version of ΔC[k], and it can create a binary reception map $M_{r}=B_{C}\left[1\right],B_{C}\left[2\right],\cdots ,B_{C}\left[k\right]$. If there is a binary sequence in $M_{r}$ corresponding to its ACK, the sensor node decides that the UAV is aware of its transmission; otherwise, it retransmits if the battery has enough energy. One can draw an analogy with a traditional modulation process: $E_{\mathrm{De}}\left(T\right)$ represents the transmitted information while $E_{\mathrm{Ab}}\left(T\right)$ is the noise.

RF-EH can be upgraded by SFC because the battery charging can use the energy from the information signal. Note that this is a disruptive approach, as traditional communication methods make it impossible for the receiver to simultaneously demodulate and recharge the battery without splitting the received energy; in other words, the battery charging process can consume completely (not partially, as discussed in Section 2.2) the energy contained in the informational signal.

### Harvesting and Consuming Energy

In order to extract the maximum available energy from the environment, the sensors keep the RF-EH circuit working even in deep sleep mode. In this way, the battery remains charged during the entire system operation. Therefore, the charge on the battery of sensor S at the instant *t* is given by
(10)$$C_{S}\left(t\right)=\left\{\begin{array}{cc} 0, & \mathrm{if}\;c_{S}\left(t\right)<0\\ C_{\max}, & \mathrm{if}\;c_{S}\left(t\right)>C_{\max}\\ c_{S}\left(t\right), & \mathrm{otherwise} \end{array}\right.,$$
with
(11)$$c_{S}\left(t\right)=E_{\mathrm{Ab}}\left(t\right)+E_{\mathrm{De}}^{\left(t\right)}-nE_{T}.$$

$E_{\mathrm{De}}^{\left(t\right)}$ is the sum of all energy packets (energy slots) sent in interval (0,t) by UAV or other sensor nodes, $E_{T}$ represents the energy wasted by transmission, and *n* is o the number of transmissions performed by the sensor S in the interval (0,t). Other energy wastes, such as consumption during battery measurement, are being neglected. One can easily add them by subtracting a factor like γt from battery charge calculations, where γ is the average consumption per second.

## 5. Case Study

**Assumption 2.** 
*For simplicity, we assume that the coverage area of a UAV, which is given by a circle of radius ρ whose center is the position of the UAV, is proportional to the energy spent on transmission. In mathematical terms, we assume that*

(12)
$$\rho =\rho_{c}E_{T},$$

*where $\rho_{c}$ is the constant of proportionality.*


We are now going to evaluate the performance of the system through simulations. For all cases studied in this work, we have η=1, $C_{\max}=100$, a carrier frequency of 400 MHz, $\sigma_{\mathrm{Ab}}=0.01$, $H_{m}=1$, $\sigma_{\mathrm{Ab}}=0.1$, Combined antenna gain of 5 dBi, $T_{E}=0.01$, and the sensors are uniformly spread over the area of interest, unless we specify the other values. The interval between the occurrences of the same event follows an Exponential distribution with mean 1/λ. The other parameters and their respective impacts on the system are discussed below. It is important to state that the scenario analyzed here provides a benchmark case where the aspects indicated in Figure 1 are covered. Our rationale is to build a simple example that can numerically demonstrate the trade-offs involved in the system performance.

As we assumed earlier, the communication between the UAV and the decision-maker takes place without errors. Furthermore, a UAV within a range of the sensor nodes always correctly receives the event signaling, and then replies with the appropriate ACK. Through these assumptions, we can characterize the event signaling success rate $R_{s}$ as a ratio proportional to the number of UAVs, the coverage area of a UAV, and the number of sensors. On the other hand, $R_{s}$ is inversely proportional to the dimension of the area of interest. Thus, by assessing $R_{s}$, we are indirectly comprehending how the $N_{s}$ and $N_{u}$ parameters impact the system’s performance. We estimate the success rate of sensor transmissions $R_{s}$ by the total number of transmissions made by the sensors divided by the total number of transmissions actually received by the UAVs.

Other metrics adopted in this work are presented as follows:The event signaling rate $R_{\mathrm{es}}$ is defined as the average probability that a sensor node transmits when an event is identified—a sensor node may fail to signal an event due to low battery power or waiting for an ACK;The event detection rate $R_{\mathrm{ed}}$ is defined as the average probability that an event is detected by the decision-maker—a relevant parameter for the controller (decision maker); in a nutshell, it represents how many events are handled by the system;Age of Information (AoI) is a relatively new metric for measuring update data, such as status or control updates. We define AoI as the elapsed time between the physical event and the time the decision-maker becomes aware of the event. It should be noted that only events received by the decision-maker are taken into account in the calculations, i.e., a possible event that was not transmitted is disregarded;The empty battery rate $R_{\mathrm{eb}}$ is defined as the probability that a sensor’s battery charge is below $E_{T}$—that is, that the sensor is unable to transmit.

### 5.1. Varying the Energy $E_{T}$

Consider a system with $N_{s}=64$, $N_{u}=5$, an area of interest measuring 40 m by 40 m, $\rho_{c}=2$, the transmission power of UAVs of $P_{\mathrm{UAV}}=100\;C_{\max}$—note that we do not deal directly with the transmitted power; instead, we represent it in a scale of the capacity of the sensors’ batteries—W=10 s and λ=1/200. The following cases are assessed by varying $E_{T}$ as {2.5,5,10,15}. For $E_{T}=2.5$, we have a consumption of 2.5% of the battery per transmission and a reduced coverage area of the UAVs (ρ=5m). On the other hand, a large coverage area (ρ=30 m) can be achieved by setting $E_{T}=15$ at the cost of high energy consumption. The intermediate values of $E_{T}$, {5.10} enable analysis regarding the trade-off between energy cost and coverage area.

Figure 6 depicts a possible configuration of sensors and UAVs. In Figure 6, we have an arrangement of sensors and UAV at a given time within the area of interest for $E_{T}=2.5$. The small blue circles are the sensors, the black × are the UAVs, and the big black circles represent the area that the UAV is covering. This hypothetical scenario highlights inadequate coverage of the areas of interest; due to an $E_{T}$ being relatively low, the coverage area of each UAV is small, leading to a significant amount of sensors being out of reach of any UAV. In this scenario, we have $R_{\mathrm{es}}=0.16$; therefore, 83% of transmission attempts are unsuccessful. A high failure rate is expected due to poor coverage quality. On the other hand, we have the event detection rate $R_{\mathrm{ed}}=0.51$. Thus, we can say that at least this first case study points out that our proposed system may be able to handle event signaling in a very effective way.

We assess the performance of the system against three other conditions: $E_{T}=5$, $E_{T}=10$, and $E_{T}=15$. Figure 7 presents the histogram of AoI for the four $E_{T}$ conditions, and the main statistics of the AoI for the evaluated cases are summarized in Table 2. By observing the behavior of the histograms in Figure 7, one can say that AoI has a distribution similar to an exponential one. The positive impact of $E_{T}$ on AoI is clear, i.e., AoI has its average, variance, and maximum value drastically reduced with the increase of $E_{T}$. Regardless of how strict the decision-maker requirement of AoI is, the proposed system can be expected to meet; as seen in Table 1, the system quickly converges to a minimum AoI—note that AoI=1, in our case, means that the delay is the smallest physically possible—with the increase of $E_{T}$. Furthermore, AoI can be further reduced by decreasing *W* due to AoI values being scaled by *W*. We will see this effect in more detail later.

Figure 8 shows the values of $R_{s}$, $R_{\mathrm{es}}$, $R_{\mathrm{ed}}$, and $R_{\mathrm{eb}}$ for the $E_{T}$ varying as {2.5,5,10,15}. $R_{s}$, $R_{\mathrm{es}}$, and $R_{\mathrm{ed}}$ have an asymptotic growth with increasing $E_{T}$. On the other hand, although we are increasing the power used in transmission by the sensor nodes, $R_{\mathrm{eb}}$ is extremely low and even negligible for $E_{T}=\left\{10,15\right\}$. A peculiar behavior of the system arises when $E_{T}=15$; we have that $R_{s}$ exceeds the others with $E_{T}=15$. In addition, $R_{s}$ reaches the value of 1 for $E_{T}=15$. One can expect that, with increasing $R_{s}$, that is, increasing the efficiency with which messages reach a UAV, $R_{\mathrm{ed}}$ also increases. This effect is visible in Figure 8, besides being clear that $R_{\mathrm{es}}=R_{\mathrm{ed}}$ for $R_{s}=1$. Such behavior can be explained by noting that every signaled event is received by the UAVs for $R_{s}=1$.

### 5.2. Varying *W*

Consider a system with $N_{s}=64$, $N_{u}=5$, an area of interest measuring 40 m by 40 m, $\rho_{c}=0.8$, the transmission power of UAVs of $P_{\mathrm{UAV}}=100C_{\max}$, $E_{T}=10$, W={1,100,200,500} and λ=1/200. The selected delays *W* somewhat cover four distinct behaviors of the system. W=1 corresponds to a no-wait system, in which sensor nodes start retransmitting as soon as possible. As briefly discussed here, SFC enables extremely effective shared access of the medium, and thus our system in the no-wait condition (W=1) does not deteriorate in performance due to collisions. On the other hand, it must be supported that the battery consumption is high for W=1. An energetically more economical behavior than W=1 is the case of W=100. In this approach, the system waits half of the mean time between events to retransmit. W=200 leads the sensor nodes to wait exactly the average time of the events, thus increasing the chance that a new event occurs before the retransmission. Finally, W=500 is certainly the condition that offers the greatest energy savings at the cost of increasing non-signaling of events. This is an extreme condition, in which the probability of retransmitting with a new event occurring is high.

Figure 9 presents the histogram of AoI for W={1,100,200,500}, and the main statistics of the AoI for the evaluated cases are summarized in Table 3. As previously stated, AoI is indeed highly dependent on *W*. Note in Table 3 that AoI is capped at 10 for W=10 and reaches 1386 for W=500. On the other hand, the histograms in Figure 9 demonstrate that in general (around 60%) the AoI maintains its minimum value (AoI = 1). Furthermore, a certain behavior of periodic peaks can be noticed in AoI for W=100 and W=200. A possible explanation for such behavior is as follows. Let the event E be monitored by a sensor. Suppose $E_{1}$ occurs at time $t_{1}$ and the sensor node transmits unsuccessfully. Then, a new transmission is performed successfully at $t_{1}+W$ in order to signal $E_{1}$. However, assume that a new event $E_{2}$ (an identical event to $E_{1}$ but at a different time) occurs in $t_{1}<t_{2}<t_{1}+W$. In this hypothetical case, the AoI is calculated by $\mathrm{AoI}=t_{1}+W-t_{2}$. In contrast, AoI=W if $t_{2}>t_{1}+W$. Therefore, we can say that, as long as the probability of $t_{2}-t_{1}>W$ is high enough, we have a predominance of AoI being a multiple of *W*. This characteristic can be noticed both in Figure 7 and Figure 9. This explanation justifies the reason for obtaining AoI of non-multiple values of *W*.

Figure 10 shows the values of $R_{s}$, $R_{\mathrm{es}}$, $R_{\mathrm{ed}}$, and $R_{\mathrm{eb}}$ for the *W* varying as {1,100,200,500}. The first point to note is the high energy consumption for retransmissions in short intervals, i.e., W=1. Because battery recharge is insufficient for W=1, the probability of battery depletion increases drastically. We can see a drop in $R_{\mathrm{es}}$ with increasing *W*; as *W* increases, the probability of a new event occurring during the wait increases. The effect of *W* on $R_{\mathrm{ed}}$ must be understood in conjunction with $R_{s}$ as follows. Let $R_{s}\left(t\right)$ be the chance of successful transmission in time *t*. Thus, since the UAVs move randomly and continuously, one can expect the correlation between $R_{s}\left(t_{1}\right)$ and $R_{s}\left(t_{2}\right)$ to be inversely proportional to $|t_{1}-t_{2}|$. The increasing curve that $R_{s}$ presents for an increase in *W* corroborates this statement. In other words, as the waiting for retransmission increases, the chances of the sensor being out of coverage again are smaller. In addition, $R_{\mathrm{ed}}$ benefits from this $R_{s}$ increase, but $R_{\mathrm{ed}}$ simultaneously deteriorates due to unsignaled events.

### 5.3. Varying $P_{\mathrm{UAV}}$

Consider a system with $N_{s}=64$, $N_{u}=5$, an area of interest measuring 40 m by 40 m, $\rho_{c}=3$, transmission power of UAVs of $P_{\mathrm{UAV}}/C_{\max}=\left\{0.5,2,5,10\right\}$, $E_{T}=10$, W=20 and λ=1/200. Our objective is now to evaluate the behavior of the battery. In this way, we vary the power of the unmodulated carrier transmitted by the UAVs for the ACK signal. We started from a relatively weak signal, $P_{\mathrm{UAV}}/C_{\max}=0.5$, to a significantly strong signal, $P_{\mathrm{UAV}}/C_{\max}=10$.

Figure 11 shows the values of $R_{s}$, $R_{\mathrm{es}}$, $R_{\mathrm{ed}}$, and $R_{\mathrm{eb}}$ for the $P_{\mathrm{UAV}}/C_{\max}$ varying as {0.5,2,5,10}. In order to highlight the effect of $R_{\mathrm{eb}}$ on $R_{\mathrm{es}}$ and $R_{\mathrm{ed}}$, we conveniently adjust the system parameters in such a way as to obtain $R_{s}=1$. It is evident that, as $R_{\mathrm{eb}}$ decreases, both $R_{\mathrm{es}}$ and $R_{\mathrm{ed}}$ increase. This behavior is explained by the fact that, as the battery remains charged enough for transmission for a longer time, the greater the probability of signaling an event. Another notable feature in Figure 11 is the equality in the values of $R_{\mathrm{es}}$ and $R_{\mathrm{ed}}$; it can be understood by keeping $R_{s}=1$.

Figure 12 presents the histogram of AoI for $P_{\mathrm{UAV}}/C_{\max}=\left\{0.5,2,5,10\right\}$, and the main statistics of the battery charge for the evaluated cases are summarized in Table 4. The evaluated scenarios demonstrate that the battery charge pdf does not have a clear pattern. The pdf has, in a way, one characteristic for $P_{\mathrm{UAV}}/C_{\max}=0.5$ and $P_{\mathrm{UAV}}/C_{\max}=2$, and another distinct one for $P_{\mathrm{UAV}}/C_{\max}=5$ and $P_{\mathrm{UAV}}/C_{\max}=10$.

## 6. Conclusions

This work addressed a remote wireless monitoring system composed of sensors and UAVs. The purpose of the system is to inform the decision-maker about events identified by the sensors. Furthermore, the sensors employ the RF-EH technique to recharge the batteries. We adopt an event transmission method called SFC that makes it possible to recharge the battery and transport information with the same energy packet. In this way, our proposal is disruptive because it can effectively and efficiently use the energy and data resources available. Simulations were performed to corroborate the advantages of our proposal in numerical terms. Through a representative case study, we discussed different aspects of the system and assessed the effect of the parameters in its performance including different metrics like error rates and AoI. Our numerical results indicated that the proposed system has the potential to be adjusted in order to meet the requirement measured in terms of AoI, which is determined by the specifications given by the end application in consideration. As a future research path, we plan to assess this framework in different realistic scenarios such as robots in an industrial plant and remote monitoring of power grids.

## Figures and Tables

**Figure 1 sensors-23-02707-f001:**
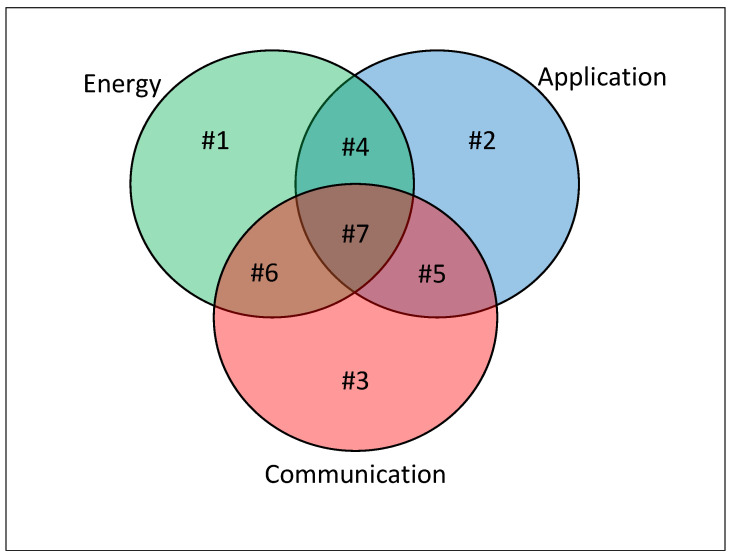
Classification of design approaches.

**Figure 2 sensors-23-02707-f002:**
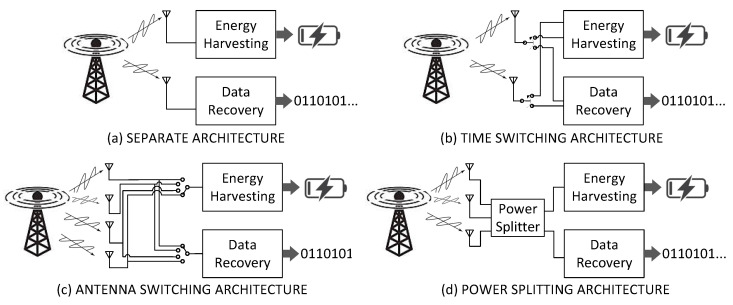
Layout of (**a**) separate, (**b**) time switching, (**c**) antenna switching, and (**d**) power splitting architecture for RF-EH.

**Figure 3 sensors-23-02707-f003:**
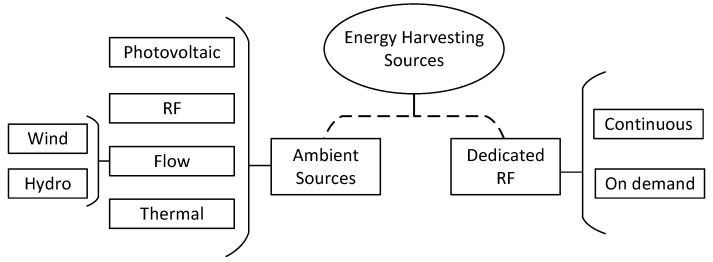
Sources to harvest energy.

**Figure 4 sensors-23-02707-f004:**
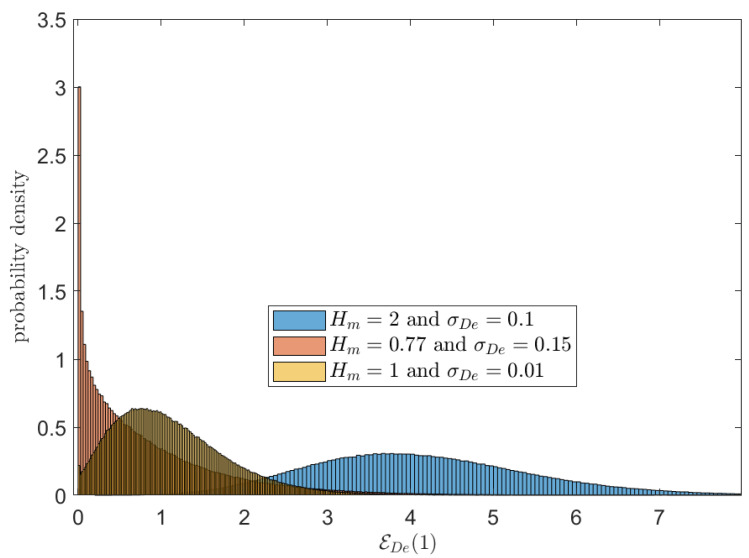
Histogram of energy harvested by RF-EH receiver $E_{\mathrm{De}}\left(T\right)$ for T=1, η=1, $A_{r}=1$ and $\sigma^{2}=0.01$.

**Figure 5 sensors-23-02707-f005:**
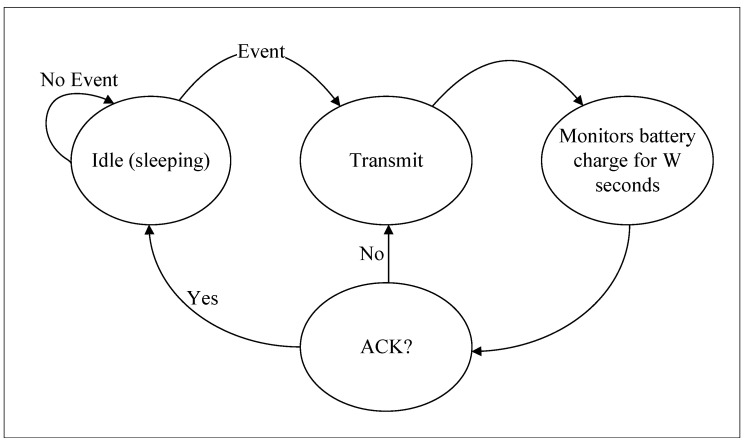
Sensor operation diagram.

**Figure 6 sensors-23-02707-f006:**
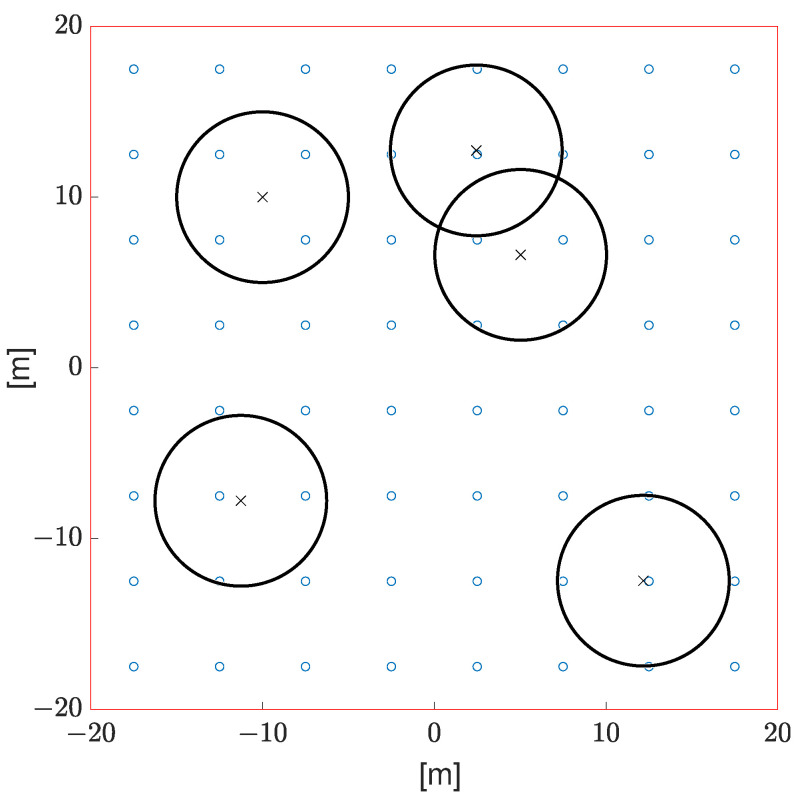
Arrangement of sensors and UAV at a given time within the area of interest for $E_{T}=2.5$. The small blue circles are the sensors, the black × are the UAVs, and the big black circles represent the area that the UAV is covering.

**Figure 7 sensors-23-02707-f007:**
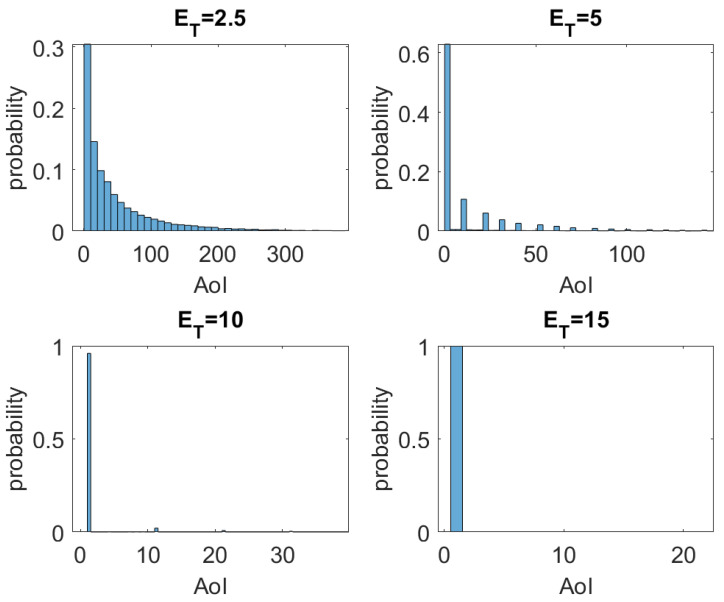
Histogram of AoI for $E_{T}=\left\{2.5,5,10,15\right\}$.

**Figure 8 sensors-23-02707-f008:**
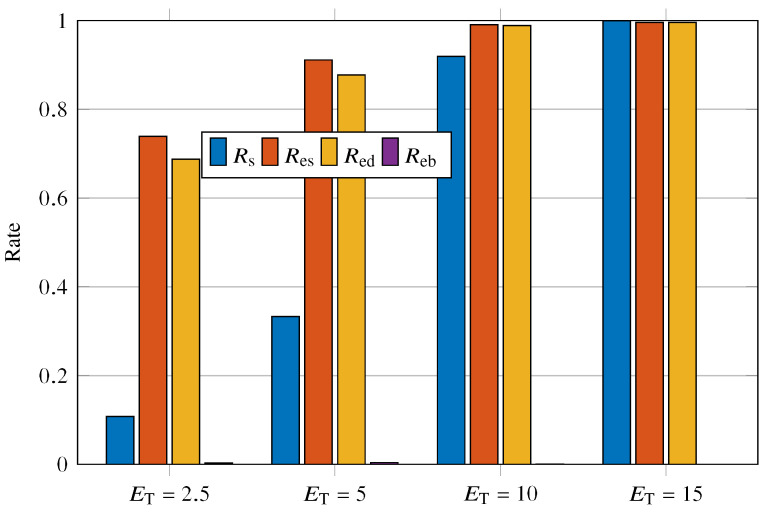
$R_{s}$, $R_{\mathrm{es}}$, $R_{\mathrm{ed}}$, and $R_{\mathrm{eb}}$ versus $E_{T}=\left\{2.5,5,10,15\right\}$.

**Figure 9 sensors-23-02707-f009:**
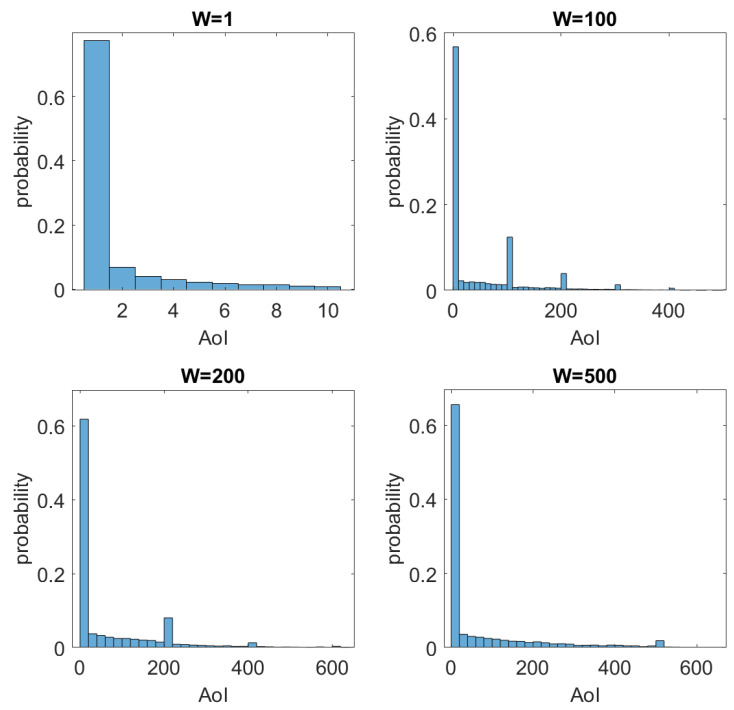
Histogram of AoI for W={1,100,200,500}.

**Figure 10 sensors-23-02707-f010:**
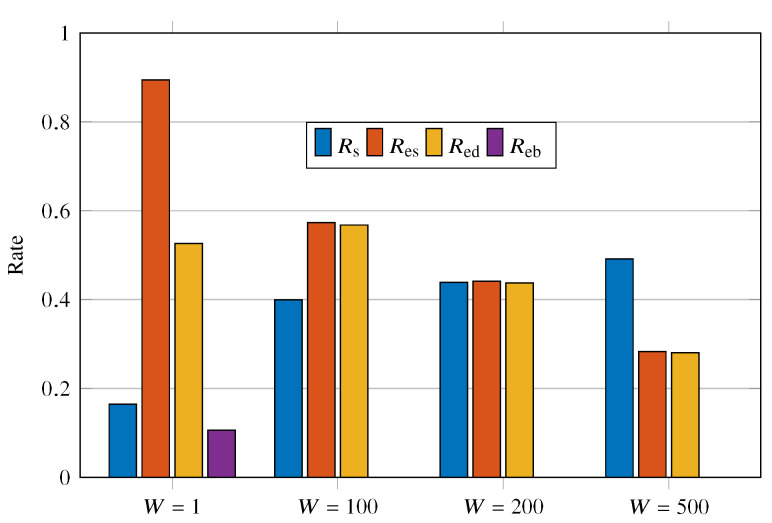
$R_{s}$, $R_{\mathrm{es}}$, $R_{\mathrm{ed}}$, and $R_{\mathrm{eb}}$ for W={1,100,200,500}.

**Figure 11 sensors-23-02707-f011:**
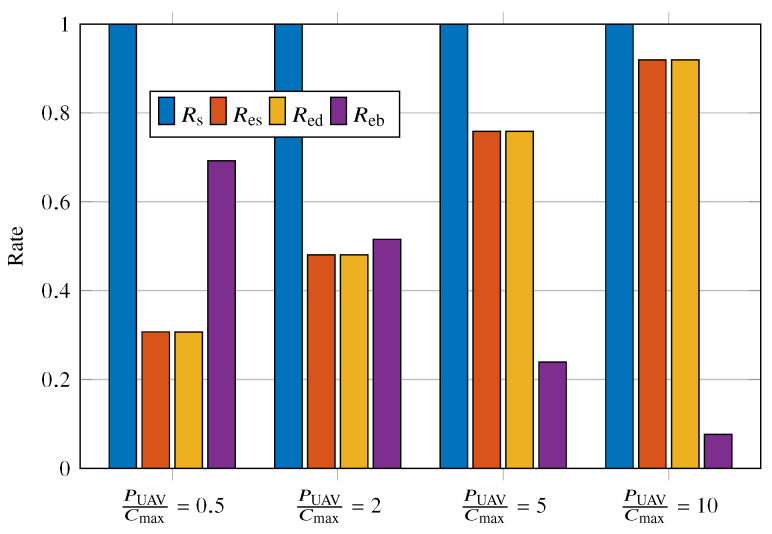
$R_{s}$, $R_{\mathrm{es}}$, $R_{\mathrm{ed}}$, and $R_{\mathrm{eb}}$ for $P_{\mathrm{UAV}}/C_{\max}=\left\{0.5,2,5,10\right\}$.

**Figure 12 sensors-23-02707-f012:**
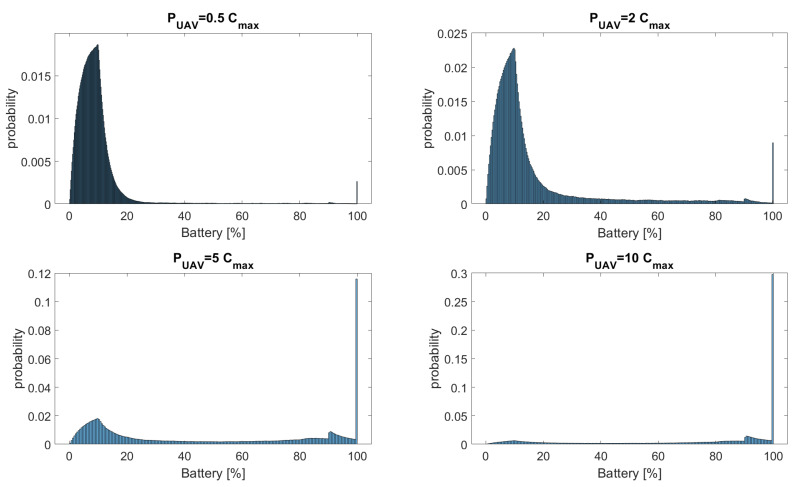
Histogram of battery charge for $P_{\mathrm{UAV}}/C_{\max}=\left\{0.5,2,5,10\right\}$.

**Table 1 sensors-23-02707-t001:** Brief description and classification of works present in the literature.

Subset	Description	Ref.
#1	Theoretical analysis of RF-DC conversion	[5,6]
#2	Event-driven data acquisition for electricity metering Automatic fault detection of sensors	[7] [8]
#3	Performance of MPSK modulation over fading channel	[9,10]
#4	Boosting quantum battery via energy harvesting	[11]
#5	Semantic-functional communications for multiuser	[12]
#6	Simultaneous wireless information and power transfer	[1,13]
#7	Communications for multiuser event transmissions and RF-EH	This work

**Table 2 sensors-23-02707-t002:** Mean, variance, and maximum AoI for the evaluated cases.

	$E_{T}=2.5$	$E_{T}=5$	$E_{T}=10$	$E_{T}=15$
Mean	46.6615	14.7353	1.7323	1.0046
Variance	4476.7869	822.9471	22.7098	0.0703
Maximum	733	253	101	21

**Table 3 sensors-23-02707-t003:** Mean, variance, and maximum AoI for the evaluated cases.

	W=1	W=100	W=200	W=500
Mean	1.7618	57.4874	68.7423	71.9253
Variance	3.2745	9122	14340	19514
Maximum	10	1334	1131	1386

**Table 4 sensors-23-02707-t004:** Mean, variance, and $R_{\mathrm{eb}}$ of battery charge for the evaluated cases.

	$P_{\mathrm{UAV}}/C_{\max}=0.5$	$P_{\mathrm{UAV}}/C_{\max}=2$	$P_{\mathrm{UAV}}/C_{\max}=5$	$P_{\mathrm{UAV}}/C_{\max}=10$
Mean	10.0205	16.5881	44.2877	70.1149
Variance	140.7149	411	1340	1160
$R_{\mathrm{eb}}$	0.68	0.52	0.24	0.08

## Data Availability

The codes used to generate the results presented in this paper are available upon request.

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
