# Peer review of "Enabling Semantic-Functional Communications for Multiuser Event Transmissions via Wireless Power Transfer"

_sensors, 2023, doi:10.3390/s23052707_

Round 1

Reviewer 1 Report

The authors present a discussion on combining wireless power transfer techniques and the semantic-function approach to consider meaningful events that require explicit use of the communications channel  to support the operation of a given cyber-physical system. They have shown through numerical results that it is possible to have an effective recharge of batteries and optimized performance by merging RF-EH and the semantic-function approach. Overall the work is of merit and interest. I have few minor suggestion to further improve the paper.

1) On what basis assumptions are chosen?

2) Please check equation 5

3) Figure 5 legends are missing

4) Proofread your paper for grammatical mistakes.

Author Response

We would like to thank the reviewer for the positive evaluation, and important constructive comments to improve the presentation of our work. We will reply here in a point-to-point fashion. The changes in the manuscript are highlighted with red font.

1) On what basis assumptions are chosen?

R: Very good point, which was indeed unclearly stated in the previous version of the work. Our objective is to propose a benchmark study that covers all three research areas in the respective field considering Energy, Communication and Application. In particular, we provided in figure 1 (added to the new version; p.2) how to jointly design the communication and RF-HE block considering the specific functionality of the system. As the complexity of the model could be high, our assumptions were determined in order to simplify the study while serving as a benchmark of the intersection showed in figure 1. In this case, we set up a simulation campaign to assess this benchmark case, and our numerical results corroborate the feasibility of this approach. However, we also understand that this model is simple, and we expect to further develop it by relaxing some assumptions, or adding more complex scenarios in future works. We have also stated our rationale in p.12, when introducing the study case, as stated below.

It is important to state that the scenario analyzed here provides a benchmark case where the aspects indicated in figure 1 are covered.
Our rationale is to build a simple example that can numerically demonstrate the trade-offs involved in the system performance.

2) Please check equation 5

R: Thanks for pointing this out; Equation 5 has now been changed.

3) Figure 5 legends are missing

R:  Thanks for this as well. We have missed this fundamental information when plotting. We have now added a legend to the axes in old figure 5 (now figure 6 in p.13) indicating that the axes are scaled in meters. Note that it is figure 6 in the new version.

4) Proofread your paper for grammatical mistakes.

R: A thorough proofread was done on the paper. We hope the changes meet expectations. All key changes are highlighted.

Reviewer 2 Report

In this work, the authors constructed a sensor network based on wireless power transmission, which solves the problem of charging the sensors while achieving information transmission for efficient energy use. Experiments were conducted to verify the feasibility of the hypothesis and the data were validated. I think the paper is acceptable, but there are still some problems, as follows.

1. In the abstract, I do not understand what the third and fourth sentences are referring to. Can you make it clearer in another way?

2. In section 4.1, what is the arrangement of ET parameters? Do you need any special designation? Please explain your rationale for designing it this way.

3. In Section 4.1, the Y-axis data in your color histogram is not labeled.

Author Response

We would like to thank the reviewer for the positive evaluation, and important constructive comments to improve the presentation of our work. We will reply here in a point-to-point fashion. The changes in the manuscript are highlighted with red font.

1. In the abstract, I do not understand what the third and fourth sentences are referring to. Can you make it clearer in another way?

R: Good point, and those sentences were indeed unclear. We rewrote both sentences (check the abstract whose main changes are highlighted). We hope that it is clearer now. In addition, a proofreading was performed on the whole paper.

2. In section 4.1, what is the arrangement of ET parameters? Do you need any special designation? Please explain your rationale for designing it this way.

R: We have added some excerpts clarifying the adopted values for Et (the old Sec. 4.1 is in this version Sec. 5.1, p. 13). The added text is transcribed below and can be found in the first paragraph of the Varying the Energy ET section.

The following cases are assessed by varying ET as {2.5, 5, 10, 15}. For ET = 2.5, we have a consumption of 2.5% of the battery per transmission and a reduced coverage area of the UAVs (ρ = 5 m ). On the other hand, a large coverage area (ρ = 30 m) can be achieved by setting ET = 15 at the cost of high energy consumption. The intermediate values of ET, {5.10}, enable analysis regarding the trade-off between energy cost and coverage area.

3. In Section 4.1, the Y-axis data in your color histogram is not labeled.

R: This is good point, and we have corrected it. We add a label to all Y-axes of the histograms (see fig. 8, p.15).

Reviewer 3 Report

The organization of this manuscript is not good. In general, problem formulation is to formulate the scenario, and then  at least  a new section is required to describe authors' method and contributions. But in this maunscript, the materials of proposed method or architecture are insufficient. It is hard for me to find the main contribution and merits compared to previous work. 

Besides, there is not comparison results with other architectures or methods. Hence, I cannot identify the value of the proposed architecture. 

Mathematical models seem too simple, without detailedly mentioning the reason or references.

Author Response

We would like to thank the reviewer for the evaluation and important criticism,  which we surely helped to improve the presentation of our work. We expect that the reviewer finds our answers and changes satisfactory. We will reply here in a point-to-point fashion. The changes in the manuscript are highlighted with red font.

1. The organization of this manuscript is not good. In general, the problem formulation is to formulate the scenario, and then at least a new section is required to describe the authors' method and contributions. But in this manuscript, the materials of the proposed method or architecture are insufficient. It is hard for me to find the main contribution and merits compared to previous work.

R: Many thanks for this comment. In the previous version, we tried to avoid being "too modular" but the result was indeed a relatively poor structure, which made difficult to see our key contributions. We have now divided the old "Problem formulation section" into two new sections. Thus, we now have one section for formulating the problem (Sec. 3, p. 8) and another containing the solution to the formulated problem (Sec. 4, p. 9). In addition, the following sentence has been added to the introduction. "The proposed system is introduced in Section 4." We have also added new sentences in the paper (highlighted parts), which we believe helps a better flow and articulation of the paper contents). We expect that those changes in the structure support the understanding of the paper's approach and results. 

2. Besides, there are no comparison results with other architectures or methods. Hence, I cannot identify the value of the proposed architecture.

R: This is a very important question, because indeed we have not explained clearly the reasons why a comparison with other methods or architectures are not presented. In order to resolve this issue, we added a figure, a table, and new explanations on how the solution contrasts the literature (p.2-3; new fig. 1 and table I). In addition, new references were also added to compare and contextualize our solution and the reason our study is unique. The added text is transcribed below.

"In addition to energy solutions, one can highlight two other aspects relevant to the design of a system: Application and Communication. For simplicity, we group all works that address concepts related to a specific function, general objective, highest communication layer (e.g., Open Systems Interconnection (OSI) model), etc., inside the Application. In a way, the Application contemplates solutions solely dealing with the primary function of operating the system. The Communication umbrella covers works whose purpose is to assess the performance of information exchange; therefore, it contains methods, techniques, and analyses that study modulation performance, coding, channel capacity, different medium access control techniques, etcetera. Finally, we can add another class that includes work related to energy consumption, named here as Energy. Energy refers to work that aims to understand and evaluate the system's energy consumption or expenditure.

An outline of how literature deals with these classes is described in fig. 1 together with table 1. Note that there are intersection regions between the subsets in fig. 1 representing works that directly address two or three aspects listed above. Table 1 is directly related to fig. 1; each subset of fig. 1 (named #1, #2, #3, etc.) is exemplified through one or more papers from the literature in table 1. Although one can extend the list of examples to subsets #1 through #6, to the best of the authors' knowledge, this is the first work to jointly consider aspects of Energy, Application, and Communication in the system design. Thus, in addition to contrasting with other works, it is not possible to draw a parallel or fairly compare our solution with other methods presented in the technical literature. In order to exemplify such a discrepancy, suppose a comparison with a Media Access Control (MAC) protocol. In this case, any clarification possibly acquired would refer exclusively to Communication, regardless of the specific method adopted (Time-Division Multiple Access (TDMA), Frequency-Division Multiple Access (FDMA), Code-Division Multiple Access (CDMA), etc.). Furthermore, one could propose a system covering the three classes (Energy, Application, and Communication) consisting of methods designed or evaluated specifically for one or two (never for all three) classes as a benchmark. This case would not be an adequate comparison either; as will become clear throughout this paper, the proposed solution jointly covers the three classes and cannot be understood as a combination of three dispersed methods."

3. Mathematical models seem too simple, without detailedly mentioning the reason or references.

R: Thanks for indicating this point, and we agree that the model is in many ways simple. However, this was our own decision because our objective with this work is to propose a benchmark study that can cover all three areas described in figure 1 (added to the new version; see above) by jointly designing communication and RF-HE block taking into account the specific functionality of the system. We have now added in Sec. 5 (p.12) a justification that indicates the reason we have selected the case study, replicated below.

It is important to state that the scenario analyzed here provides a benchmark case where the aspects indicated in figure 1 are covered.
Our rationale is to build a simple example that can numerically demonstrate the trade-offs involved in the system performance.

Round 2

Reviewer 3 Report

Authors give a detailed  revision regarding reviewer's comment, I agree to accept this version.